# Risk of severe influenza infection in women with a history of pregnancy complications: A longitudinal cohort study

Amira Amer[1,2], Aimina Ayoub[2,3], Émilie Brousseau[2,3], Nathalie Auger[1,2,3,4] *

**1** Department of Social and Preventive Medicine, School of Public Health, University of Montreal, Montreal, Quebec, Canada, **2** University of Montreal Hospital Research Centre, Montreal, Quebec, Canada, **3** Institut national de Santé Publique du Québec, Montreal, Quebec, Canada, **4** Department of Epidemiology, Biostatistics and Occupational Health, McGill University, Montreal, Quebec, Canada

* nathalie.auger@inspq.qc.ca

## Abstract

**Data Availability Statement:** The underlying data for this study are owned by the Quebec Health Ministry and are available upon request from the Institut de la statistique du Québec (ISQ)

### Background

Risk factors for influenza complications in women are poorly understood. We examined the association between pregnancy outcomes and risk of influenza hospitalization up to three decades later.

### Methods

We analyzed a cohort of 1,421,531 pregnant women who delivered in Quebec, Canada between 1989 and 2021. Patients were followed over time beginning at the first delivery. The main exposure measures included obstetric complications such as preeclampsia, gestational diabetes, and preterm birth. The main outcome was influenza hospitalization up to 32 years later. We used adjusted Cox regression models to estimate hazard ratios (HR) and 95% confidence intervals (CI) for the association between obstetric complications and risk of influenza hospitalization following pregnancy.

### Results

A total of 4,016 women were hospitalized for influenza during 32 years of follow-up. Influenza hospitalization was more frequent among women with pregnancy complications than women without complications (18.0 vs 14.1 per 100,000 person-years). Compared with no pregnancy complication, women with gestational diabetes (HR 1.48, 95% CI 1.30–1.69), preeclampsia (HR 1.45, 95% CI 1.28–1.65), placental abruption (HR 1.36, 95% CI 1.12–1.66), preterm birth (HR 1.40, 95% CI 1.27–1.55), cesarean section (HR 1.22, 95% CI 1.13–1.31), and severe maternal morbidity (HR 1.43, 95% CI 1.22–1.68) had a greater risk of influenza hospitalization later in life. These pregnancy outcomes were associated with severe influenza infections requiring critical care.

repository. Access to the data is subject to standard access procedures. For more information, please visit: https://www.stat.gouv.qc.ca/recherche/#/accueil.

**Funding:** This work was supported by the Canadian Institutes of Health Research (grant number PJT-162300) and Fonds de recherche du Québec-Santé (grant number 296785). the funders had no role in study design, data collection and analysis, decision to publish, or preparation of the manuscript.

**Competing interests:** The authors have declared that no competing interests exist.

## Conclusions

Women with pregnancy complications have an elevated risk of severe influenza complications later in life and have potential to benefit from seasonal vaccination to prevent influenza hospitalization.

## Introduction

Pregnant women are at risk of severe influenza infections during gestation [1], but the long-term risk of influenza morbidity after a complicated pregnancy has not been studied. Seasonal influenza affects up to 11% of pregnant women annually [1, 2], and accounts for a considerable number of hospitalizations and deaths during gestation [3]. Pregnant women are susceptible to more severe influenza infections owing to physiologic changes that affect cardiopulmonary function and cell-mediated immunity [4], with data suggesting that gestational influenza infection is associated with preterm birth and stillbirth [5, 6]. For this reason, pregnant women are considered a high risk group and are included in influenza vaccination guidelines [7]. However, the possibility that adverse birth outcomes may be an indicator of future risk of influenza morbidity has not been considered.

A growing number of studies suggest that adverse pregnancy outcomes may be predictors of infectious morbidity later on [8, 9]. In an analysis of 1.8 million women from the US, preterm birth and postpartum hemorrhage were associated with the risk of sepsis up to 9 months later [8]. A study of over 1 million women from Canada found that preterm birth and gestational diabetes were associated with two times the risk of developing necrotizing fasciitis several years following pregnancy [9]. Pregnancy complications are also linked with chronic morbidity later on. Outcomes such as gestational diabetes and preeclampsia are associated with an increased risk cardiovascular disease up to 30 years after pregnancy [10]. Some of these associations are thought to relate to inflammatory or endothelial pathways that increase the risk of both pregnancy complications and chronic disease [11, 12]. However, the relationship between pregnancy and subsequent influenza morbidity has not been investigated. A better understanding of reproductive risk factors for influenza morbidity may be informative for vaccination guidelines. We examined the association between pregnancy outcomes and risk of influenza hospitalization up to three decades later.

## Materials and methods

### Study design and population

We carried out a longitudinal cohort study of 1,421,531 women enrolled in the Maintenance and Use of Data for the Study of Hospital Clientele registry in Quebec, Canada. The registry has diagnostic and intervention data for every pregnant patient admitted to hospital for delivery between 1989 and 2021 [13]. The registry includes most pregnant women in Quebec as healthcare is publicly funded and delivery care is generally provided in hospital. The data also contain hospitalizations that occurred after pregnancy, including admissions for influenza infection.

We extracted all pregnant women who delivered a live born or stillborn infant in hospital during the study period. We began follow-up at the first pregnancy and, using unique patient identifiers, identified all subsequent influenza hospitalizations in the period between delivery

and March 31, 2021. We excluded women who died at delivery or lacked a unique identifier, because we could not track these patients over time.

## Pregnancy-related exposures

The main exposure measures included six pregnancy complications selected on the basis of prevalence and potential to be associated with influenza morbidity: gestational diabetes mellitus, preeclampsia, placental abruption, preterm delivery (<37 weeks of gestation), cesarean section, and severe maternal morbidity. Severe maternal morbidity comprised life-threatening complications such as severe preeclampsia or eclampsia, embolism, shock, disseminated intravascular coagulation, acute renal failure, and other severe complications following the definition of the Public Health Agency of Canada [14]. Secondary exposures of interest included placenta previa, placenta accreta, other placental complications (infarction, retention, membranous placenta), antepartum or postpartum hemorrhage, amniotic fluid volume imbalance (polyhydramnios, oligohydramnios), chorioamnionitis, premature rupture of membranes, multiple birth, stillbirth, low birth weight (<2500 g), and fetal macrosomia (≥4000 g).

We identified pregnancy complications using diagnostic codes in the 9th and 10th revisions of the International Classification of Diseases (ICD), and procedure codes in the Canadian Classification of Diagnostic, Therapeutic, and Surgical Procedures and Canadian Classification of Health Interventions [9]. Gestational age was collected in completed weeks and birth weight in grams.

## Influenza hospitalization

The main outcome was hospitalization for influenza during follow-up. We determined the severity of infection by identifying patients with influenza who were admitted to an intensive care unit, required intubation, developed adult respiratory distress syndrome, or had superimposed pneumonia. We identified patients with these outcomes using diagnostic and intervention codes (S1 Table).

## Covariates

We used a directed acyclic graph to identify factors that could potentially influence the risk of pregnancy complications and influenza morbidity (S1 Fig). Covariates included age at delivery (<25, 25–34, ≥35 years), metabolic conditions including obesity, preexisting diabetes, preexisting hypertension, and dyslipidemia (yes, no), autoimmune disorders such as rheumatologic conditions, vasculitis, celiac disease, Graves' disease, Guillain-Barré syndrome, autoimmune thyroiditis, and myasthenia gravis (yes, no) [9], alcohol, tobacco and other substance use disorders (yes, no), socioeconomic disadvantage (yes, no, unknown), rurality (yes, no, unknown), and time period (1989–1998, 1999–2008, 2009–2021). Socioeconomic disadvantage was measured as the bottommost quintile of a neighborhood-level index that accounts for education level, income, and employment [15].

## Statistical analysis

We calculated the rate of influenza hospitalization per 100,000 person-years. We computed the incidence per 1,000 women after 32 years of follow-up using the cumulative incidence function, accounting for deaths as a competing event [16]. Using Cox proportional hazards regression models, we estimated hazard ratios (HR) and 95% confidence intervals (CI) for the association between pregnancy complications and later influenza hospitalization. We examined associations with gestational diabetes, preeclampsia, placental abruption, preterm birth,

cesarean section, and severe maternal morbidity in the first, second, or any later pregnancy. For secondary pregnancy exposures, we focused on events at the first delivery. We adjusted the models for age, metabolic conditions, autoimmune disorders, substance use disorders, socio-economic disadvantage, rurality, and time period, and expressed the time scale as the number of days between the first delivery and the date of influenza hospitalization, death, or end of study. We analyzed death as a competing event and censored women who were never admitted for an influenza infection before the end of the study period. We evaluated the proportional hazards assumption using log (–log survival) curves.

In an additional set of adjusted regression models, we estimated the association with severe influenza infection, defined as influenza requiring intensive care admission or intubation, and influenza with superimposed pneumonia or adult respiratory distress syndrome. We performed the analysis in SAS version 9.4 (SAS Institute Inc., Cary, NC, USA).

### Ethics approval

We obtained an ethical waiver from the University of Montreal Hospital Centre institutional review board. The data were anonymized and informed consent was not required.

### Results

Among 1,421,531 pregnant women who delivered between 1989 and 2021, 4,016 women later developed influenza infections requiring hospitalization during 25,055,739 person-years of follow-up (Table 1). The overall hospitalization rate for influenza was 16.0 per 100,000 person-years (95% CI 15.5–16.5). Rates were higher for women with gestational diabetes, preeclampsia, placental abruption, preterm delivery, cesarean section, and severe maternal morbidity. Women with metabolic disorders, autoimmune disorders, substance use disorders, and socio-economic disadvantage had particularly elevated rates of influenza hospitalization.

Pregnancy complications were associated with influenza hospitalization later in life (Table 2). Gestational diabetes (HR 1.48, 95% CI 1.30–1.69), preeclampsia (HR 1.45, 95% CI 1.28–1.65), placental abruption (HR 1.36, 95% CI 1.12–1.66), preterm delivery (HR 1.40, 95% CI 1.27–1.55), cesarean section (HR 1.22, 95% CI 1.13–1.31), and severe maternal morbidity (HR 1.43, 95% CI 1.22–1.68) were all associated with an increased risk of subsequent influenza hospitalization compared with no pregnancy complication. Antepartum hemorrhage (HR 1.27, 95% CI 1.06–1.52), stillbirth (HR 1.57, 95% CI 1.10–2.23), and low birth weight (HR 1.65, 95% CI 1.49–1.82) were also associated with later influenza hospitalization.

Patients with pregnancy complications had an elevated cumulative incidence of influenza hospitalization after 32 years of follow-up (Fig 1). The cumulative incidence was highest for severe maternal morbidity with a rate of 9.3 influenza admissions per 1,000 women, followed by gestational diabetes (8.9 per 1,000), preeclampsia (8.5 per 1,000), preterm birth (7.5 per 1,000), cesarean section (7.1 per 1,000), and placental abruption (7.0 per 1,000). The cumulative incidence did not surpass 5.5 per 1,000 in women without these complications. For most pregnancy complications, the increase in incidence started to become apparent 5 years following delivery and continued to grow during the remainder of follow-up compared with no pregnancy complication.

Risk of influenza hospitalization varied according to the number of complicated pregnancies (Table 3). Women with two or more pregnancies affected by severe maternal morbidity (HR 2.67, 95% CI 1.68–4.25), preterm birth (HR 2.25, 95% CI 1.89–2.69), gestational diabetes (HR 1.99, 95% CI 1.67–2.36), preeclampsia (HR 1.88, 95% CI 1.41–2.52), and cesarean section (HR 1.37, 95% CI 1.24–1.51) had the greatest risk of influenza hospitalization compared with

**Table 1. Influenza hospitalization rate according to characteristics at first pregnancy.**

| | Total no. women | No. hospitalized for influenza | Influenza hospitalization rate per 100,000 person-years (95% confidence interval) |
|---|---|---|---|
| Gestational diabetes | 75,875 | 247 | 24.8 (21.9–28.1) |
| Preeclampsia | 65,471 | 257 | 24.1 (21.3–27.3) |
| Placental abruption | 30,471 | 99 | 21.8 (17.9–26.5) |
| Preterm delivery | 104,851 | 405 | 22.5 (20.4–24.8) |
| Cesarean section | 318,364 | 972 | 18.9 (17.8–20.2) |
| Severe maternal morbidity | 39,871 | 160 | 23.8 (20.4–27.8) |
| Age, years | | | |
| <25 | 368,359 | 1,428 | 20.5 (19.5–21.6) |
| 25–34 | 890,474 | 2,209 | 14.2 (13.6–14.8) |
| ≥35 | 162,698 | 379 | 15.3 (13.8–16.9) |
| Metabolic disorder[a] | 56,859 | 277 | 31.8 (28.3–35.8) |
| Autoimmune disorder[b] | 9,196 | 48 | 46.9 (35.3–62.2) |
| Substance use disorder[c] | 23,796 | 97 | 25.7 (21.1–31.4) |
| Socioeconomic disadvantage | | | |
| Yes | 270,214 | 916 | 19.9 (18.6–21.2) |
| No | 1,040,056 | 2,651 | 14.8 (14.2–15.4) |
| Rurality | | | |
| Yes | 238,144 | 800 | 18.6 (17.4–19.9) |
| No | 1,114,676 | 2,928 | 15.4 (14.8–16.0) |
| Time period | | | |
| 1989–1999 | 607,354 | 2,500 | 15.0 (14.4–15.6) |
| 2000–2010 | 419,151 | 946 | 14.8 (13.9–15.7) |
| 2011–2021 | 395,026 | 570 | 28.4 (26.1–30.8) |
| Total | 1,421,531 | 4,016 | 16.0 (15.5–16.5) |

[a]Obesity, preexisting diabetes, preexisting hypertension, dyslipidemia.

[b]Rheumatologic disorder, vasculitis, celiac disease, Graves' disease, Guillain-Barré syndrome, autoimmune thyroiditis, myasthenia gravis, other autoimmune disorders.

[c]Tobacco, alcohol, and drug use disorders.

no complication. The risk was not as elevated for women with only one complicated pregnancy.

Pregnancy complications were strongly associated with severe influenza infection (Table 4). Women with gestational diabetes were more likely to have influenza infections that required intensive care admission (HR 2.11, 95% CI 1.54–2.90) or intubation (HR 2.46, 95% CI 1.46–4.17), or that progressed to adult respiratory distress syndrome (HR 2.68, 95% CI 1.24–5.79). All other pregnancy complications were also associated with influenza requiring intensive care, although to a lesser extent. Cesarean section was associated with influenza requiring intubation (HR 1.57, CI 1.13–2.18) and influenza-related adult respiratory distress syndrome (HR 1.74, 95% CI 1.05–2.89). Placental abruption, severe maternal morbidity, preterm birth, and preeclampsia were associated with the risk of superimposed pneumonia.

## Discussion

In this longitudinal cohort study of 1.4 million women with 25 million person-years of follow-up, pregnancy complications were associated with an increased risk of influenza hospitalization up to three decades following delivery. Gestational diabetes, preeclampsia, placental

**Table 2. Association between pregnancy complications and future risk of influenza hospitalization.**

| | Influenza hospitalization rate per 100,000 person-years | | Hazard ratio (95% confidence interval)[a] | |
| --- | --- | --- | --- | --- |
| | Exposed | Unexposed | Unadjusted | Adjusted |
| Gestational diabetes | 24.8 | 15.7 | 1.60 (1.40–1.82) | 1.48 (1.30–1.69) |
| Preeclampsia | 24.1 | 15.7 | 1.57 (1.39–1.78) | 1.45 (1.28–1.65) |
| Placental abruption | 21.8 | 15.9 | 1.40 (1.14–1.70) | 1.36 (1.12–1.66) |
| Preterm delivery | 22.5 | 15.5 | 1.46 (1.32–1.62) | 1.40 (1.27–1.55) |
| Cesarean section | 18.9 | 15.3 | 1.25 (1.16–1.34) | 1.22 (1.13–1.31) |
| Severe maternal morbidity | 23.8 | 15.8 | 1.53 (1.31–1.79) | 1.43 (1.22–1.68) |
| Placenta previa | 14.0 | 16.0 | 0.87 (0.52–1.47) | 0.89 (0.52–1.50) |
| Placenta accreta | 18.3 | 16.0 | 1.14 (0.91–1.42) | 1.13 (0.90–1.41) |
| Other placental complication[b] | 14.4 | 16.1 | 0.90 (0.73–1.12) | 0.92 (0.74–1.14) |
| Antepartum hemorrhage | 20.0 | 15.9 | 1.28 (1.06–1.53) | 1.27 (1.06–1.52) |
| Postpartum hemorrhage | 18.9 | 15.9 | 1.20 (1.06–1.36) | 1.12 (0.99–1.27) |
| Polyhydramnios | 19.4 | 16.0 | 1.22 (0.86–1.72) | 1.09 (0.77–1.54) |
| Oligohydramnios | 18.3 | 16.0 | 1.18 (0.92–1.51) | 1.07 (0.83–1.36) |
| Chorioamnionitis | 16.2 | 16.0 | 1.04 (0.90–1.19) | 0.98 (0.86–1.13) |
| Premature rupture of membranes | 16.2 | 16.0 | 1.03 (0.91–1.15) | 0.97 (0.86–1.09) |
| Multiple birth | 13.1 | 16.1 | 0.82 (0.61–1.11) | 0.82 (0.60–1.10) |
| Stillbirth | 26.0 | 16.0 | 1.62 (1.14–2.31) | 1.57 (1.10–2.23) |
| Low birth weight, <2500 g | 26.4 | 15.3 | 1.72 (1.56–1.91) | 1.65 (1.49–1.82) |
| Macrosomia, ≥4000 g | 16.2 | 16.0 | 1.01 (0.91–1.13) | 1.03 (0.92–1.15) |
| Any complication | 18.0 | 14.1 | 1.29 (1.22–1.38) | 1.25 (1.17–1.33) |

[a]Hazard ratio for pregnancy complication vs. no complication, adjusted for age, metabolic disorders, autoimmune disorders, substance use disorders, socioeconomic disadvantage, rurality, and time period.
[b]Placenta infarction, retention, and membranous placenta.

abruption, preterm birth, cesarean section, and severe maternal morbidity were all moderately associated with the risk of influenza hospitalization later in life. Risks were also elevated for women who had two or more pregnancies with these complications. Women with adverse pregnancy outcomes were more likely to have a severe influenza infection that required intensive care or intubation, progressed to adult respiratory distress syndrome, or presented with superimposed pneumonia. Our findings suggest that pregnancy complications may be moderate indicators of susceptibility to severe influenza infection in the years following delivery. Women with complicated pregnancies may potentially benefit from influenza vaccination and be considered a high-risk group in vaccination guidelines to prevent severe influenza morbidity.

To our knowledge, previous studies have not examined the long-term risk of influenza morbidity in women with pregnancy complications. The literature has focused on outcomes of influenza during pregnancy only, not the period after pregnancy [1, 2, 5, 6]. A prospective cohort study from the U.K. found that pregnant women who were hospitalized for influenza during gestation were 21 times more likely to require intensive care and 1.4 times more likely to have a cesarean section compared with women who were admitted for delivery without influenza [1]. Up to 7.7 pregnant women per 10,000 with influenza require hospitalization, and intensive care unit admission rates can be as high as 6.8 per 10,000 pregnancies [2]. An analysis of 1.4 million pregnant women found that influenza hospitalization increased the risk of preterm birth and low birth weight by at least 25% compared with no influenza [5]. A recent

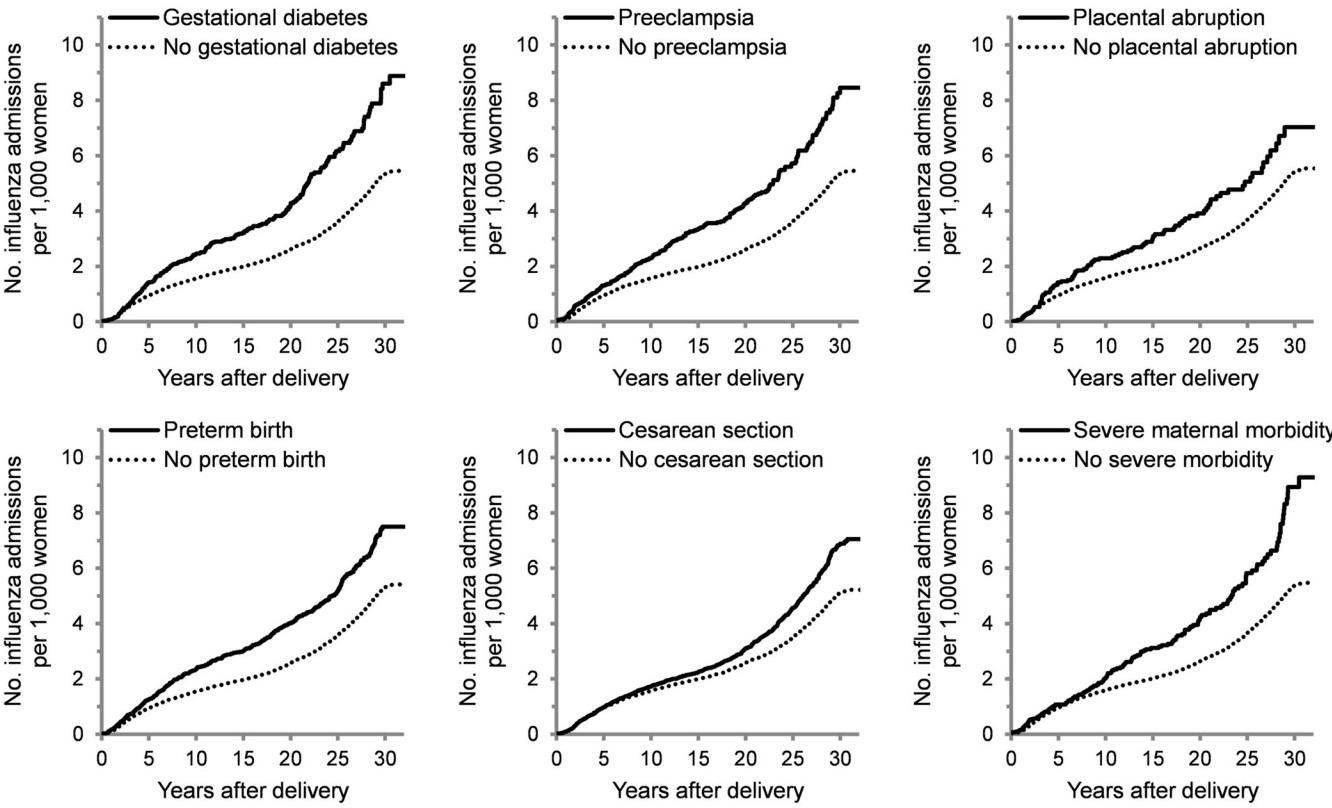

**Fig 1. Cumulative incidence of influenza hospitalization over time in women with pregnancy complications.**

meta-analysis of over 2.3 million patients from 17 different cohorts reported that influenza during pregnancy was associated with three times the risk of stillbirth compared with no infection [6]. While it is clear that influenza during pregnancy increases the risk of adverse obstetric outcomes, it is unclear whether obstetric outcomes are associated with influenza morbidity after pregnancy.

Pregnancy complications are known to be associated with other types of infectious morbidity well past delivery [8, 9, 17]. An analysis of 1.9 million women from the U.S. reported that cesarean section, preterm birth, and postpartum hemorrhage were associated with two to five times the risk of sepsis hospitalization at 9 months postpartum [8]. In a longitudinal cohort of 1.3 million women from Canada, complications such as gestational diabetes, severe maternal morbidity, cesarean section, and preterm birth were associated with a two times greater risk of developing necrotizing fasciitis up to 3 decades later [9]. Similarly, a cohort study of 1.1 million women found that preeclampsia was associated with the development of Guillain-Barré syndrome up to 25 years after delivery [17]. Guillain-Barré syndrome is a rare neurologic complication of influenza or other infectious diseases [17]. These studies therefore support the possibility that pregnancy complications may be indicators of risk for severe influenza infections that develop later in life [8, 9, 17].

Among different pregnancy outcomes in our data, gestational diabetes stood out as the complication that was the greatest risk factor for severe influenza infections requiring intensive care or intubation, or progressing to adult respiratory distress syndrome. Gestational diabetes is often associated with metabolic disorders and cardiovascular disease that develop later in life [10, 18]. A meta-analysis of 20 studies found that women with gestational diabetes were 10

**Table 3. Association between number of pregnancy complications and risk of influenza hospitalization among multiparous women.**

| | Total no. women | No. hospitalized for influenza | Influenza hospitalization rate per 100,000 person-years (95% confidence interval) | Hazard ratio (95% confidence interval)[a] |
|---|---|---|---|---|
| No. pregnancies with gestational diabetes | | | | |
| 0 | 738,184 | 2,307 | 16.8 (16.1–17.5) | Reference |
| 1 | 62,692 | 230 | 22.3 (19.6–25.3) | 1.23 (1.07–1.40) |
| ≥2 | 25,221 | 141 | 38.2 (32.4–45.1) | 1.99 (1.67–2.36) |
| No. pregnancies with preeclampsia | | | | |
| 0 | 772,673 | 2,434 | 17.2 (16.5–17.9) | Reference |
| 1 | 45,708 | 197 | 24.0 (20.9–27.6) | 1.35 (1.16–1.56) |
| ≥2 | 7,716 | 47 | 33.6 (25.3–44.8) | 1.88 (1.41–2.52) |
| No. pregnancies with placental abruption | | | | |
| 0 | 787,505 | 2,520 | 17.4 (16.7–18.1) | Reference |
| 1 | 36,702 | 149 | 23.9 (20.4–28.1) | 1.31 (1.11–1.55) |
| ≥2 | 1,890 | 9 | 27.4 (14.3–52.7) | 1.51 (0.78–2.89) |
| No. preterm deliveries | | | | |
| 0 | 718,267 | 2,115 | 16.1 (15.4–16.8) | Reference |
| 1 | 90,381 | 435 | 26.1 (23.8–28.7) | 1.57 (1.42–1.74) |
| ≥2 | 17,449 | 128 | 38.5 (32.4–45.7) | 2.25 (1.89–2.69) |
| No. cesarean sections | | | | |
| 0 | 608,337 | 1,847 | 16.4 (15.6–17.1) | Reference |
| 1 | 83,520 | 309 | 19.6 (17.5–21.9) | 1.21 (1.07–1.36) |
| ≥2 | 134,240 | 522 | 23.2 (21.3–25.3) | 1.37 (1.24–1.51) |
| No. pregnancies with severe maternal morbidity | | | | |
| 0 | 787,097 | 2,479 | 17.2 (16.6–17.9) | Reference |
| 1 | 37,037 | 181 | 26.1 (22.5–30.2) | 1.48 (1.27–1.72) |
| ≥2 | 1,963 | 18 | 48.8 (30.7–77.4) | 2.67 (1.68–4.25) |

[a]Adjusted for age, metabolic disorders, autoimmune disorders, substance use disorders, socioeconomic disadvantage, rurality, and time period.

**Table 4. Association between pregnancy complications and severe influenza infection.**

| | Hazard ratio (95% confidence interval)[a] | | | | |
|---|---|---|---|---|---|
| | Intensive care unit admission (N = 516) | Intubation (N = 177) | Adult respiratory distress syndrome (N = 77) | Superimposed pneumonia (N = 309) | Influenza admission without severe complication (N = 3,352) |
| Gestational diabetes | 2.11 (1.54–2.90) | 2.46 (1.46–4.17) | 2.68 (1.24–5.79) | 1.39 (0.85–2.28) | 1.42 (1.22–1.64) |
| Preeclampsia | 1.69 (1.22–2.35) | 1.32 (0.70–2.50) | 0.60 (0.14–2.48) | 1.97 (1.32–2.93) | 1.39 (1.21–1.60) |
| Placental abruption | 1.77 (1.08–2.93) | 2.06 (0.91–4.67) | 2.36 (0.75–7.47) | 2.04 (1.12–3.74) | 1.31 (1.05–1.64) |
| Preterm delivery | 1.52 (1.15–2.00) | 1.49 (0.92–2.40) | 0.89 (0.35–2.22) | 1.96 (1.41–2.71) | 1.35 (1.20–1.51) |
| Cesarean section | 1.35 (1.10–1.64) | 1.57 (1.13–2.18) | 1.74 (1.05–2.89) | 1.31 (1.01–1.70) | 1.18 (1.09–1.28) |
| Severe maternal morbidity | 1.88 (1.28–2.78) | 1.89 (0.97–3.71) | 1.46 (0.46–4.66) | 1.76 (1.05–2.95) | 1.35 (1.13–1.62) |

[a]Hazard ratio for pregnancy complication vs. no complication, adjusted for age, metabolic disorders, autoimmune disorders, substance use disorders, socioeconomic disadvantage, rurality, and time period.

times more likely to develop type 2 diabetes compared with normoglycemic pregnant women [18]. Hyperglycemia is known to impede infectious disease defense mechanisms [19]. Individuals with type 2 diabetes are up to 10 times more likely to be hospitalized for influenza [20], and over 4 times more likely to require intensive care than nondiabetic hospital patients [21]. As a result, it is recommended that individuals with diabetes be vaccinated annually against influenza [21]. Our findings suggest that gestational diabetes could be an additional indication for vaccination in influenza guidelines.

Preeclampsia, placental abruption, and severe maternal morbidity were also associated with the risk of influenza hospitalization. Women with these complications had a 36% to 45% greater risk of influenza hospitalization over time, particularly severe influenza infections with superimposed pneumonia or intensive care unit admission. Severe preeclampsia, eclampsia, and placental abruption are all components of severe maternal morbidity [14, 22]. Placental disorders such as preeclampsia and placental abruption are known to be associated with endothelial cell dysfunction [12]. Endothelial dysfunction has also been observed in severe influenza infections as a consequence of excessive cytokine production [23]. Placental disorders associated with endothelial dysfunction are frequently included in guidelines for prevention of non-infectious morbidities following pregnancy [24]. The U.S. and the U.K. recommend long-term monitoring of patients with a history of preeclampsia to prevent adverse cardiovascular outcomes [24]. All these factors suggest that placental disorders could also be considered in guidelines for annual influenza vaccination.

Preterm birth was also associated with an increased risk of influenza hospitalization. Reasons why preterm birth may be associated with severe influenza infections are more difficult to assess, as preterm birth has a multitude of causes. Preterm birth can be either spontaneous or medically indicated [25]. Spontaneous preterm birth is thought to be multifactorial and potentially related to subclinical systemic inflammation brought on by infection or stress [25]. Both infection and stress are associated with an increased presence of inflammatory biomarkers, including tumor necrosis factor alpha and C-reactive protein [25]. Activation of inflammatory pathways could be present after delivery at a subclinical level, compromising the host's immunity to pathogens such as influenza. Between 30–40% of preterm births are medically-indicated due to pregnancy complications such as preeclampsia [26]. Endothelial dysfunction in such patients could therefore contribute to the association of preterm birth with subsequent influenza hospitalization [27]. Medically indicated preterm birth also frequently clusters with cesarean delivery [26], which may explain why cesarean section was associated with influenza hospitalization in our data.

In our study, multiparous women with two or more complicated pregnancies were considerably more likely to be hospitalized for influenza than multiparous women with one or no complicated pregnancy. Recurrent gestational diabetes, preeclampsia, preterm birth, and severe maternal morbidity were associated with two to three times the risk of influenza hospitalization, compared with no complicated pregnancies. As immune dysregulation and chronic inflammation are believed to be common to gestational diabetes, preeclampsia, preterm birth, and severe maternal morbidity [22, 24, 25, 28], our findings suggest that immune-mediated pathways may underlie the risk of severe influenza infection in patients with recurrent pregnancy complications.

This study had a large sample size and a lengthy follow-up period, but there are nevertheless limitations. We cannot rule out the possibility of misclassification of exposures and outcomes, although this problem is likely negligible as medical administrative data in Quebec are rigorously validated [13]. Residual confounding is possible because we could not account for medical conditions diagnosed in outpatient settings. We could not account for changes in influenza vaccination over time, although any changes are likely to have affected patients with and

without a history of pregnancy complications equally. Some characteristics, such as obesity and substance use, may be underreported in the data. We did not have information on women who delivered outside of hospital, although these patients account for less than 1% of pregnancies in the population. We could not analyze mild influenza infections and infections that led to hospitalization outside of Quebec. Further research is needed to determine the extent to which the number of pregnancy complications affects the risk of influenza hospitalization later in life. The data are representative of a multicultural Canadian population with publicly funded healthcare. Generalizability to other settings is to be determined.

In this cohort study of 1.4 million pregnant women spanning three decades, women with gestational diabetes, severe maternal morbidity, preeclampsia, placental abruption, preterm birth, and cesarean section were all at risk of hospitalization for severe influenza infections later in life, especially infections requiring critical care. Current influenza vaccination guidelines focus primarily on patients who are pregnant or patients with morbidities such as type 1 or 2 diabetes and pulmonary conditions [7]. More consideration should be given to including women with a past history of pregnancy complications as an additional priority group during annual influenza vaccination campaigns.

## Supporting information

**S1 Fig. Directed acyclic graph (DAG) for pregnancy complications and influenza hospitalization.** The DAG was prepared using DAGitty v3.1.
(TIF)

**S1 Table. Diagnostic codes for influenza and related complications.**
(DOCX)

## Author Contributions

**Conceptualization:** Amira Amer, Aimina Ayoub, Émilie Brousseau, Nathalie Auger.

**Formal analysis:** Amira Amer, Aimina Ayoub.

**Funding acquisition:** Nathalie Auger.

**Methodology:** Amira Amer, Aimina Ayoub, Émilie Brousseau, Nathalie Auger.

**Project administration:** Nathalie Auger.

**Resources:** Nathalie Auger.

**Supervision:** Aimina Ayoub, Émilie Brousseau, Nathalie Auger.

**Visualization:** Aimina Ayoub, Émilie Brousseau, Nathalie Auger.

**Writing – original draft:** Amira Amer, Émilie Brousseau, Nathalie Auger.

**Writing – review & editing:** Aimina Ayoub.

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
