## [Decision Letter · Decision Letter 0]

6 Sep 2024

PONE-D-23-39142Risk of severe influenza infection in women with a history of pregnancy complications: a longitudinal cohort studyPLOS ONE

Dear Dr. Auger,

Thank you for submitting your manuscript to PLOS ONE. After careful consideration, we feel that it has merit but does not fully meet PLOS ONE’s publication criteria as it currently stands. Therefore, we invite you to submit a revised version of the manuscript that addresses the points raised during the review process – specially in the Methods’ section.

We look forward to receiving your revised manuscript.

Kind regards,

Fernanda Penido Matozinhos, Ph.D

Academic Editor

PLOS ONE

Journal Requirements:

Did you know that depositing data in a repository is associated with up to a 25% citation advantage (https://doi.org/10.1371/journal.pone.0230416)? If you’ve not already done so, consider depositing your raw data in a repository to ensure your work is read, appreciated and cited by the largest possible audience. You’ll also earn an Accessible Data icon on your published paper if you deposit your data in any participating repository (https://plos.org/open-science/open-data/#accessible-data).

"This work was supported by the Canadian Institutes of Health Research (grant number PJT-162300) and Fonds de recherche du Québec-Santé (grant number 296785)."

4. Please note that funding information should not appear in the Acknowledgments section or other areas of your manuscript. We will only publish funding information present in the Funding Statement section of the online submission form. Please remove any funding-related text from the manuscript.

5. We note that you have indicated that there are restrictions to data sharing for this study. For studies involving human research participant data or other sensitive data, we encourage authors to share de-identified or anonymized data. However, when data cannot be publicly shared for ethical reasons, we allow authors to make their data sets available upon request. For information on unacceptable data access restrictions, please see http://journals.plos.org/plosone/s/data-availability#loc-unacceptable-data-access-restrictions. 

**Additional Editor Comments:**

Dear Authors,

Thank you for submitting your manuscript to PLOS ONE!

After careful consideration, we feel that the goal of this study is relevant and it has technical rigor, but does not fully meet PLOS ONE’s publication criteria as it currently stands. Therefore, we invite you to submit a revised version of the manuscript that addresses the points raised during the review process – specially in the Methods’ section.

Kind regards,

Reviewers' comments:

Reviewer's Responses to Questions

**Comments to the Author**

1. Is the manuscript technically sound, and do the data support the conclusions?

Reviewer #1: Yes

Reviewer #2: Yes

2. Has the statistical analysis been performed appropriately and rigorously? 

Reviewer #1: Yes

Reviewer #2: Yes

3. Have the authors made all data underlying the findings in their manuscript fully available?

Reviewer #1: Yes

Reviewer #2: Yes

4. Is the manuscript presented in an intelligible fashion and written in standard English?

Reviewer #1: Yes

Reviewer #2: Yes

5. Review Comments to the Author

Reviewer #1: An interesting work with a significant number, but that includes different social responses to the disease throughout the time period.It would be important to clarify that over such a long period of time the approach strategies vary equally and are not the same.

Reviewer #2: It is my pleasure to review this study. This study examined the association of obstetric complications with influenza hospitalization for up to 32 years among more than 1 million pregnant women. In general, this is a well-conducted and well-written paper. I have a few comments that the authors may consider.

1. Covariate selection: the authors may consider using DAGs

2. There is no mention of how to calculate cumulative incidence. The K-M method without considering competing events may lead to a biased estimate of cumulative incidence.

3. How about assumptions of proportional hazard? What approach has been used to evaluate this assumptions?

4. As one may experience multiple complications during one pregnancy, the authors may consider the dose-relationship between the number of pregnancy complications and risks of influenza.

6. PLOS authors have the option to publish the peer review history of their article (what does this mean?). If published, this will include your full peer review and any attached files.

Reviewer #1: No

Reviewer #2: **Yes: **Yongfu Yu, School of Public Health, Fudan University, China

---

## [Author Response · Author response to Decision Letter 0]

17 Oct 2024

Fernanda Penido Matozinhos, PhD

Academic Editor

PLOS ONE

17 September 2024

Ref: PONE-D-23-39142R1

Dear Dr. Matozinhos,

Thank you for inviting us to prepare a revised version of our original research article, “Risk of severe influenza infection in women with a history of pregnancy complications: a longitudinal cohort study”. We wish to thank the Editors and Reviewers for their time and helpful comments. We have provided a detailed response to each comment and incorporated the recommendations in the manuscript.

We confirm that the funders had no role in study design, data collection and analysis, decision to publish, or preparation of the manuscript.

We provide the following data availability statement: The underlying data used in the study are owned by the Quebec Health Ministry and are available upon request from the Institut de la statistique repository following standard access procedures, https://www.stat.gouv.qc.ca/recherche/#/accueil.

We hope you will be pleased with the revised manuscript and remain available for additional corrections. Thank you for considering our research for publication in PLOS ONE.

Kind regards,

Nathalie Auger MD

RESPONSE TO REVIEWS

Editor Comments:

E.1. Please ensure that your manuscript meets PLOS ONE's style requirements, including those for file naming. The PLOS ONE style templates can be found at 

Response: We ensured that we followed style requirements.

E.2. Note from Emily Chenette, Editor in Chief of PLOS ONE, and Iain Hrynaszkiewicz, Director of Open Research Solutions at PLOS: 

Did you know that depositing data in a repository is associated with up to a 25% citation advantage (https://doi.org/10.1371/journal.pone.0230416)? If you’ve not already done so, consider depositing your raw data in a repository to ensure your work is read, appreciated and cited by the largest possible audience. You’ll also earn an Accessible Data icon on your published paper if you deposit your data in any participating repository (https://plos.org/open-science/open-data/#accessible-data).

Response: We confirm that the raw data are available from the Institut de la statistique du Québec repository and can be accessed upon request following standard procedures (https://www.stat.gouv.qc.ca/recherche/#/accueil).

E.3. Thank you for stating the following financial disclosure: 

"This work was supported by the Canadian Institutes of Health Research (grant number PJT-162300) and Fonds de recherche du Québec-Santé (grant number 296785)."

Response: We added the role of the funders in the cover letter: “the funders had no role in study design, data collection and analysis, decision to publish, or preparation of the manuscript”.

E.4. Please note that funding information should not appear in the Acknowledgments section or other areas of your manuscript. We will only publish funding information present in the Funding Statement section of the online submission form. Please remove any funding-related text from the manuscript.

Response: We deleted the funding information from the manuscript.

E.5. We note that you have indicated that there are restrictions to data sharing for this study. For studies involving human research participant data or other sensitive data, we encourage authors to share de-identified or anonymized data. However, when data cannot be publicly shared for ethical reasons, we allow authors to make their data sets available upon request. For information on unacceptable data access restrictions, please see http://journals.plos.org/plosone/s/data-availability#loc-unacceptable-data-access-restrictions. 

Response: We confirm that due to legal restrictions we cannot share the data. The data are owned by a third party, the Quebec Health Ministry. Any researcher wishing to access the data must make a formal request following standard procedures. We clarified the data availability statement as follows: “The underlying data used in the study are owned by the Quebec Health Ministry and are available upon request from the Institut de la statistique repository following standard access procedures, https://www.stat.gouv.qc.ca/recherche/#/accueil”.

E.6. Please review your reference list to ensure that it is complete and correct. If you have cited papers that have been retracted, please include the rationale for doing so in the manuscript text, or remove these references and replace them with relevant current references. Any changes to the reference list should be mentioned in the rebuttal letter that accompanies your revised manuscript. If you need to cite a retracted article, indicate the article’s retracted status in the References list and also include a citation and full reference for the retraction notice.

Response: We confirm that the reference list is complete and correct.

E.7. Thank you for submitting your manuscript to PLOS ONE!

After careful consideration, we feel that the goal of this study is relevant and it has technical rigor, but does not fully meet PLOS ONE’s publication criteria as it currently stands. Therefore, we invite you to submit a revised version of the manuscript that addresses the points raised during the review process – specially in the Methods’ section.

Response: We thank the Editors and Reviewers and have carefully incorporated their recommendations. We hope you will be pleased with the revised manuscript.

Reviewer #1:

R1.1. An interesting work with a significant number, but that includes different social responses to the disease throughout the time period. It would be important to clarify that over such a long period of time the approach strategies vary equally and are not the same.

Response: We thank the Reviewer. We confirm that patients admitted towards the end of the study may be more likely to be vaccinated. However, we have no reason to suspect that influenza vaccination levels differ between patients with and without pregnancy complications. We added the limitation that “We could not account for changes in influenza vaccination over time, although any changes are likely to have affected patients with and without a history of pregnancy complications equally” (page 17, lines 2-4).

Reviewer #2:

It is my pleasure to review this study. This study examined the association of obstetric complications with influenza hospitalization for up to 32 years among more than 1 million pregnant women. In general, this is a well-conducted and well-written paper. I have a few comments that the authors may consider.

R2.1. Covariate selection: the authors may consider using DAGs

Response: We thank the Reviewer and added a DAG in the supplement. We added that “We used a directed acyclic graph to identify factors that could potentially influence the risk of pregnancy complications and influenza morbidity (S1 Fig)” (page 6, lines 5-6).

R2.2. There is no mention of how to calculate cumulative incidence. The K-M method without considering competing events may lead to a biased estimate of cumulative incidence.

Response: We used the cumulative incidence function to account for competing events (Lin 2012). We clarified that “We computed the incidence per 1,000 women after 32 years of follow-up using the cumulative incidence function, accounting for deaths as a competing event [16]” (page 6, lines 17-19).

Lin G, et al. Analyzing survival data with competing risks using SAS® software. Orlando, FL: SAS Institute Inc.; 2012.

R2.3. How about assumptions of proportional hazard? What approach has been used to evaluate this assumptions?

Response: We added that “We evaluated the proportional hazards assumption using log (−log survival) curves” (page 7, lines 6-7).

R2.4. As one may experience multiple complications during one pregnancy, the authors may consider the dose-relationship between the number of pregnancy complications and risks of influenza.

Response: We agree with the Reviewer that patients could have more than one complication, including more than one pregnancy with multiple complications. As it was challenging to address multiple pregnancies with multiple complications in a time-to-event analysis, we added the recommendation that “Further research is needed to determine the extent to which the number of pregnancy complications affects the risk of influenza hospitalization later in life” (page 17, lines 8-9).

---

## [Decision Letter · Decision Letter 1]

29 Oct 2024

Risk of severe influenza infection in women with a history of pregnancy complications: a longitudinal cohort study

PONE-D-23-39142R1

Dear Nathalie Auger,

We’re pleased to inform you that your manuscript has been judged scientifically suitable for publication and will be formally accepted for publication once it meets all outstanding technical requirements.

Kind regards,

Fernanda Penido Matozinhos, Ph.D

Academic Editor

PLOS ONE

Additional Editor Comments (optional):

Dear Author,

After careful consideration, I feel the manuscript explores a very important topic. The modifications in the text made the manuscript come to a satisfying result. I recommend its publication.

Kind regards, Fernanda Penido.

Reviewers' comments:

Reviewer's Responses to Questions

**Comments to the Author**

1. If the authors have adequately addressed your comments raised in a previous round of review and you feel that this manuscript is now acceptable for publication, you may indicate that here to bypass the “Comments to the Author” section, enter your conflict of interest statement in the “Confidential to Editor” section, and submit your "Accept" recommendation.

Reviewer #1: All comments have been addressed

Reviewer #2: All comments have been addressed

2. Is the manuscript technically sound, and do the data support the conclusions?

Reviewer #1: Yes

Reviewer #2: Yes

3. Has the statistical analysis been performed appropriately and rigorously? 

Reviewer #1: Yes

Reviewer #2: Yes

4. Have the authors made all data underlying the findings in their manuscript fully available?

Reviewer #1: Yes

Reviewer #2: No

5. Is the manuscript presented in an intelligible fashion and written in standard English?

Reviewer #1: Yes

Reviewer #2: Yes

6. Review Comments to the Author

Reviewer #1: The modifications made to the manuscript are adequate and methodologically valid. The text is acceptable and complies with the journal's standards. A very nice text.

Reviewer #2: (No Response)

7. PLOS authors have the option to publish the peer review history of their article (what does this mean?). If published, this will include your full peer review and any attached files.

Reviewer #1: No

Reviewer #2: **Yes: **Yongfu Yu, School of Public Health, Fudan University, China

---

## [Editor Report · Acceptance letter]

4 Nov 2024

PONE-D-23-39142R1 

PLOS ONE

Dear Dr. Auger, 

I'm pleased to inform you that your manuscript has been deemed suitable for publication in PLOS ONE. Congratulations! Your manuscript is now being handed over to our production team.

Kind regards, 

on behalf of

Dr. Fernanda Penido Matozinhos 

Academic Editor

PLOS ONE